# The Dynamic Relationship among Bank Credit, House Prices and Carbon Dioxide Emissions in China

**DOI:** 10.3390/ijerph191610428

**Published:** 2022-08-21

**Authors:** Guangyang Chen, Kai Dong, Shaonan Wang, Xiuli Du, Ronghua Zhou, Zhongwei Yang

**Affiliations:** 1Business School, Nanjing Xiaozhuang University, Nanjing 211171, China; 2Department of Agriculture and Applied Economics, University of Georgia, Conner Hall, 147 Cedar St., Athens, GA 30602, USA; 3School of Finance, Nanjing University of Finance and Economics, Nanjing 210023, China; 4School of Social Sciences and Humanities, University College Sedaya International, Cheras 56000, Malaysia

**Keywords:** bank credit, house price, carbon dioxide emissions, dynamic relationship

## Abstract

This paper explores the dynamic relationship among bank credit, house prices and carbon dioxide emissions in China by systematically analyzing related data from January 2000 to December 2019 with the help of the time-varying parameter vector autoregression with stochastic volatility (TVP-SV-VAR) model and the Bayesian DCC-GARCH model. Empirical results show the expansion of bank credit significantly drives up house prices and increases carbon dioxide emissions in mosttimes. The rise in house prices inhibits the expansion of bank credit but increases carbon dioxide emissions and aggravates environment pollution, and that the increase in carbon dioxide is helpful to stimulate bank credit expansion and house price rise. In addition, bank credit and house prices are most relevant, followed by bank credit and carbon dioxide emissions, then by house prices and carbon dioxide emissions. Therefore, we believe that in order to stabilize skyrocketing house prices, restrain carbon dioxide emissions, and secure a stable and healthy macro-economy, the government should strengthen management of bank credit, and effectively control its total volume.

## 1. Introduction

For a long time, bank credit has been considered an important factor affecting the smooth development of China’s macroeconomics. Especially after the 2008 financial crisis, in order to avoid the risk of a hard landing of the economy, the Chinese Central Government issued a great quantity of currency. Against this background, commercial banks sharply expand credit, which in order to drive social investment, expand domestic consumption, curb the downward trend in economic growth, and finally help to achieve steady and rapid macroeconomic development. However, the marginal utility of expanded bank credit has gradually weakened in recent years, and its negative effects increasingly stand out. What needs to be emphasized is that, under the “new normal”, China’s economic transformation has entered a critical period, and structural reforms for economic growth have entered a painful period. At present, the benefits of the real economy are generally low, therefore a large amount of credit funds are entangled among financial institutions for arbitrage by idling, which leads to a rapid rise in financial leverage, cumulates financial risk speedily and ultimately initiates the violent turbulence in the financial system. Recent huge volatility in stock and bond markets has sounded the alarm. The Central Government noticed the seriousness of this problem. Therefore, at the 19th CPC National Congress, they proposed to “strengthen control on monetary policy and macroprudential policy” to improve the macroprudential framework and maintain stability in the financial sector. At the Central Economic Work Conference held in December 2017, they made it clear to “control money supply”. Since then, the growth rate of bank credit has declined, with a clearer trend of credit tightening.

Currently, China’s economic growth is slowing down and corporate profitability is declining, but house prices are high and carbon dioxide emissions have skyrocketed. China carried out a comprehensive market-oriented reform of real estate in 1998, since then, house prices have maintained a long-term trend of rapid rise. By the end of 2019, the extent of the cumulative increasement had outstripped 220% (the figure was calculated by authors based on relevant data from the official website of National Bureau of Statistics of China). As the expected returns of most other investment channels and assets are obviously lower than the real estate market, a huge amount of credit funds has flowed into the housing market in all kinds of forms or through various channels. At present, real estate is not only the most common mortgage deposit, but also as the subject matter for a large number of financial products in China’s financial market. Therefore, it is not difficult to find that the real estate market is increasingly closely linked with the financial system and macro-economy with the passage of time. The historical experience has repeatedly proved to us that the large fluctuations in property prices often act as the trigger of financial systemic crises and negative shocks of macroeconomics. Additionally, the sustained and rapid rise of house prices will not only have a profound and long-term impact on economic activities, but also on energy consumption and environmental quality. Some studies have pointed out that the development of the real estate market or a great demand for housing have profoundly influenced carbon dioxide emissions. Every one percentage point increase in the growth of the real estate market will lead to an increase of 0.010% in the short term and 0.009% in the long term for the carbon level in Turkey [1]. Specifically, the rise in house prices leads to the appreciation of real estate, improves the wealth of households with houses, reduces the cost of credit, helps stimulate the consumption and investment behavior of the household sector, thus stimulating energy consumption (more prominent for those developing economies that are highly dependent on traditional energy), production, investment expenditure and economic growth, and ultimately leads to the increase of carbon dioxide emissions.

As we all know, so far, China’s economy has maintained rapid growth for only 40 years and created outstanding economic achievements. However, for a long time the extensive development mode characterized by factor driving has led to a lot of energy consumption and carbon emissions, which is not conducive to economic and social sustainability. In recent years, due to the transformation and upgrading of the internal economy and rapid changes in external trade environment, the growth rate of carbon dioxide rises sharply. The rapid, sustained and long-term growth of carbon dioxide emissions greatly affects the stability of the sustainable economic development in the long run. In September 2020, China clearly made a promise to the whole world about its goal of carbon peak and carbon neutrality in 2030 and 2060. In the 14th Five Year Plan of China, it clearly proposed the policy recommendations and corresponding supporting measures for making every effort to achieve the “double carbon target” (peaking carbon emissions by 2030 and achieving carbon neutrality by 2060). This is the basic strategy to promote the construction of a low-carbon and zero-carbon society in China. It is a major strategic arrangement and strategic task for China to achieve high-quality development during the high-quality transformation period of economic development. It is also a strategic measure for China to improve the influence of the international community and strive to build a community of human civilization. Some documents have clearly pointed out that the exponential escalation in house prices and the sharp increase in carbon dioxide emissions are inextricably linked to bank credit [2,3]. The surge in bank credit and the resulting flooding of liquidity are often seen as early warning signs or potential risks of poor economic performance. The overdose of credit supply and liquidity flooding usually bring about skyrocketing house prices, increased energy consumption inevitably or drove carbon emissions at a high speed, and results in the economy overheating, which will inevitably endanger the entire financial system and the macro-economy in the future.

The most important issues to be discussed in this manuscript include: What is the connection among bank credit, house prices and carbon dioxide emissions? What is the degree of the relation among them? Answers to these questions can help practically achieve the goal of “strengthen macroprudence and Promote high-quality economic development”, and maintain the stable and healthy operation of the macro-economy under the new normal. The principal substance of the rest of this paper includes: literature review in Section 2, which systematically combs the research literature related to the theme; description of the TVP-SV-VAR model and the DCC-GARCH model in Section 3; description of empirical results in Section 4 on the analysis of the dynamic relationship degree among bank credit, house prices and carbon dioxide emissions, and the degree of dynamic relation among the three elements; conclusions and recommendations in Section 5.

## 2. Literature Review

For now, the research about the relevance, correlation or interaction among bank credit, house prices and carbon dioxide emissions can be differentiated as three categories in general: those on the impact of bank credit on house prices, on the impact of bank credit on carbon dioxide emissions, and on the impact of house prices on carbon dioxide emissions.

### 2.1. The Impact of Bank Credit on House Prices

At present, a large part of academic literature has pointed out that credit conditions are an important force driving house price fluctuations. The real estate cycle is caused by cheaper and easier access to credit and the subsequent credit supply contraction [4,5,6]. Lamont and Stein (1999) pointed out there is a positive relationship between a city’s house prices and availability of credit [7]. Edelstein and Paul (2000) believed that the soaring financial leverage is probably the principal consideration for the formation of the house price foam, especially pointing out that the main reasons for the formation of Japan’s real estate foam are the excessive savings in the banking system caused by net international capital inflows and the resulting low loan costs [8]. Geanakoplos and zame (2014) considered that the growth rate of the leverage ratio (private sector credit/GDP) in the financial system is a key factor driving the rapid rise of asset prices, including house price [9]. Gete and Reher (2016) pointed out that, compared to an economy with lagging financial development, real estate prices are more sensitive to changes in financial variables in countries with relatively developed financial markets, and the fluctuations in property prices will magnify macroeconomic tremors through the “financial accelerator” channel of household lending behavior [10]. Duca et al. (2021) proved based on cross-country panel data that the availability of credit and credit model have a significant impact on house prices [11]. Ping and Chen (2004) and Zhang et al. (2006) believed the increase of housing credit will promote the rise of housing prices remarkably, they indicated that property loans and house prices were positively correlated in China, and that government-supported bank credit significantly promotes the rise of house prices [12,13]. Through empirical research, Liang and Gao (2007) revealed that the scale of bank credit has a greater impact on house prices in eastern and western China, but a smaller impact on house prices in central China [14]. Tan and Wang (2011) combined the multivariate GARCH model and dynamic stochastic general equilibrium (DSGE) model and discovered that there is a significant mutual promotion between bank credit and house prices [15]. Wang et al. (2013) and Jia et al. (2014) used China’s credit and house price data for empirical analysis, the results pointed out that credit expansion caused by loose monetary policy and liquidity flooding are reliable explanations for the rapid increase of house prices [16,17]. Yu and Huang (2015) investigated the regional heterogeneity of loose monetary policy with the global vector autoregressive model, and found that compared with central and western cities, money supply has a greater positive impact on house prices in eastern cities [18]. Bologna et al. (2022) and Mian and Sufi (2022), respectively, pointed out that the release of quotas (although funds were relaxed for mortgage supply) and speculation were the key channels for the expansion of bank credit supply to affect the real estate market. The study found that after deregulation, banks that had previously been strictly restricted would release more liquidity, which would lead to mortgage expansion and accelerated the rise of house prices. This impact was achieved by increasing housing demand and relaxing the financial constraints of borrowers. In addition, mortgage securitization was an important reason for the surge in the scale of mortgage loans. Driven by a small number of speculators, the real estate transaction volume and real estate prices in non-core areas of the city were more likely to rise sharply. These areas have amplified the degree of prosperity and depression of the real estate market, which was in line with the expectation of belief heterogeneity [19,20]. Some other scholars emphasized the important impact of loose monetary policy, cheaper credit and expectations of house prices from the perspective of home buyers’ characteristics and believed that the credit costs are the most important factor driving house price fluctuations [21,22,23].

However, not all studies believe that bank credit can significantly affect house prices. Park et al. (2010) used the data of South Korea from 1988 to 2006 and found that the impact of reducing bank credit on house prices in policy target areas is very weak, but it is very effective in non-policy target areas [24]. Glaeser et al. (2012) analyzed the realistic characteristics of the U.S. real estate market cycle and pointed out that, during the recession of the real estate market, every percentage decline in bank credit can lead to a decline in real estate prices of 0.3% to 0.6%. In sharp contrast, during the boom of the real estate market, changes in bank credit did not play a strong role in driving up house prices [25]. Wei and Chen (2017) showed that house prices are very sensitive to changes in mortgage leverage, whose impact on house prices has a non-linear dual threshold effect. In particular, the dual threshold effect in eastern China appeared earlier than in central and western regions [26]. Kaplan et al. (2020) believed that the loose credit condition was not enough to explain the rapid rise in house prices. Compared with abundant liquidity or low bank credit costs, the impact of heterogeneity expectations on house prices was more obvious and important [27].

### 2.2. The Impact of Bank Credit on Carbon Dioxide Emissions

There is a wealth of research on the factors affecting carbon emissions. For example, economic growth, industrial structure, energy endowment, urbanization, energy efficiency, technological innovation and financial development are all considered to be the key factors driving carbon emissions. However, the research of the mechanism, degree and characteristics of financial development’s effect on carbon emissions is still clearly controversial in academia, including bank credit on carbon emissions. Some studies from the literature have concluded that bank credit plays a positive and active role in the process of carbon emission reduction. For example, some scholars argued that bank credit can reduce energy consumption and carbon dioxide emissions by promoting technological progress, upgrading industrial structure and improving energy efficiency [28,29]. Abbasi and Riaz (2016) utilized the total credit and private sector credit to represent the financial variables, and then employed the ARDL method, Error Correction Model (ECM), Granger causality and augmented VAR approach to explore the influence of financial variables on carbon emissions in a small emerging economy, and the research conclusion showed that the carbon dioxide emission reduction effect of financial variables was not significant in the early stage of economic development, especially in the early stage of financial development, but with the deepening of financial development and financial liberalization, the carbon emission reduction effect of financial variables would gradually become prominent [30]. Zhao and Yang (2020) utilized the principal component analysis to explore the mechanism and influence degree of financial development on carbon dioxide emissions with China’s provincial panel data. The empirical results showed that as a proxy variable for the financial development indicator, one standard increment in provincial financial development level would lead to a drop in carbon emissions by 4–5% on average, demonstrating that regional financial development was conducive to reducing carbon dioxide emissions, which means the inhibitory effects of financial development on CO_2_ emissions were conspicuous [31]. Many studies have shown that green credit (in this article, green credit refers to the financial institutions in the banking industry using the interest rate lever to regulate the flow of credit funds and realize the “green allocation” of funds) has an effect on energy structure energy intensity, energy efficiency and resource allocations, but only a small part of literature directly investigated the impact of green credit on carbon emissions. Jiang et al. (2020) claimed that green credit is crucial to achieving carbon peak and carbon neutrality [32]. Kang et al. (2020) and An et al. (2021) used the GCF model from the perspective of the supply chain and showed that green credit is helpful to manufacturers for reaching the goals of carbon emission reduction [33,34].

However, some other studies have come to a completely different conclusion from the above, that bank credit is likely to increase carbon dioxide emissions and deteriorate environmental quality [35,36]. Zhang (2011) conducted the ratio of loan to GDP as the proxy variable of financial development and used the cointegration test and variance decomposition method to analyze the relationship between financial development and carbon dioxide emissions. The research results showed that financial development is an important driving force of carbon dioxide emissions, and there is a long-term stable relationship between them [37]. A similar result is also verified through taking advantage of credit to the private sector as the alternating quantity of financial development [35]. Haseeb et al. (2018) and Ahmad et al. (2018) utilized the Granger causality test, ARDL and error correction model (ECM) methods to explore the impact of financial development on carbon dioxide emissions. They found that there was an obvious self-reinforcing cycle effect between financial development and carbon dioxide emissions [38,39]. Shahbaz et al. (2021) used the relevant data of G7 countries from 1870 to 2014, and the empirical analysis showed that the impact of financial development on carbon emissions has been nonlinear for a long time, and takes on different shapes in different countries. Specifically, in Canada, Japan and the United States, the impact of financial development on carbon dioxide emissions presents an “M” shape, France, Italy and the United Kingdom present an inverted “N” shape, and Germany presents a “W” shape [40]. Additionally, Boutabba (2014) and Kim et al. (2021) also obtained similar conclusions in recent studies [41,42].

### 2.3. The Impact of House Prices on Carbon Dioxide Emissions

Over the past two decades, the strong housing demand in China has significantly pushed up the levels of house prices and profoundly affected the macro-economy activities. Subsequently, as an important indicator of the development level of the real estate market, house prices are bound to have an impact on the amount, scale and structure of energy consumption or its closely related carbon dioxide emissions that cannot be ignored. However, there are only very few examples from the literature that have paid attention to the impact of the real estate market on carbon dioxide emissions or environmental quality [43,44,45]. Glaeser and Kahn (2010) used the data of the United States to quantitatively analyze the carbon dioxide emissions of travel modes, household electricity and heating, and new houses in different regions. The study found that Texas and Oklahoma had the highest carbon dioxide emissions, while California had the lowest. In addition, the study also found that there is a long-term stable negative correlation between land-use regulations and carbon dioxide emissions. Urban emissions were usually much lower than suburbs and the gap between cities and suburbs was particularly large in older areas such as New York. Climate conditions, urban population and density were very important for carbon externality pricing [46]. Zhang et al. (2018) discovered that the extensive use of green standards, green design and environmental protection technologies in the construction industry could effectively reduce carbon dioxide emissions and improve environmental quality, which was regarded as necessary measures or key steps to achieve high quality economic development. In addition to technological development, the economic feasibility also played a crucial role in promoting the design, research and development, promotion and use of green buildings [47]. Qashou et al. (2022) showed that the development of the real estate market is not conducive to Turkey’s carbon emission reduction. The growth of the real estate market or the rise of house prices by 1% would drive Turkey’s carbon dioxide emissions to rise by 0.010% in the short term and 0.009% in the long term. Therefore, Turkey should design a new strategy for the sustainable real estate market and improve the environmental quality by supporting green investment. In addition, the study also found that the development of renewable energy will largely lead to carbon dioxide emission reduction, whether in the short term or long term, while the increase in real income would accelerate the emission of most exhaust gases, including carbon dioxide in the long term [1]. Additionally, some studies have highlighted the importance of balancing costs and benefits in the implementation of green infrastructure [48,49].

Some other studies have pointed out that the energy-saving real estate investments or labels have a certain impact on house prices. For example, Dinan and Miranowski (1989) accepted the view that improving efficiency saving investments capitalized into housing prices. The study found a negative correlation among energy consumption, air quality, environmental cleanliness and real estate transaction price in Des Moines, Iowa [50]. With the help of a hedonic price model, Gilmer (1989) applied the economic search model and the Minnesota energy-saving housing sample to analyze the ability of the household energy rating system to help buyers quickly and accurately identify the correct pricing of property. Our case study found that the benefits of the search were small but positive [51]. Eichholtz et al. (2010, 2013) argued that the rent of those properties with the “Energy Star” label (indicating that the property is among the top quarter of the most energy-efficient properties) was 2–3% higher than ordinary properties, while the selling price was 13–16% higher. Further empirical analysis discovered that the actual energy consumption was closely related to the premium of environmentally friendly real estate, which indicated that most tenants, investors, buyers and sellers in the commercial real estate sector have capitalized energy conservation in investment decisions [52,53].

In summary, lots of existing research literature tends to put emphasis on the impact of bank credit on house prices or carbon dioxide emissions, and of house prices on carbon dioxide emissions, but fails to reach consistent conclusions. The mentioned manuscripts in the literature review above have not combined bank credit, house prices and carbon dioxide emissions in the same analysis framework or measurement model to analyze the mutual influence or dynamic relationship. Therefore, this study systematically analyzes the interaction and dynamic relation among them with the help of the TVP-SV-VAR model and the Bayesian DCC-GARCH model, and puts forward some referential and constructive suggestions on how to maintain healthy, long-term and sustainable economic development in China, in order to perform the useful replenishment and prolongation of existing research.

## 3. Model Description

Primiceri (2005)proposed a vector autoregressive model (TVP-SV-VAR) with time-varying parameters and random fluctuations based on the SVAR model [54]. Engle (2002) generalized the constant condition correlation (CCC) GARCH model to the dynamic condition correlation (DCC) GARCH model [55]. The above two models are often used to handle financial data sequences with noisy, nonlinear and dynamic characteristics. This article mainly discusses the dynamic relationship among bank credit, house prices and carbon dioxide emissions. The so-called dynamic relevance is manifested as the influence of a variable change on others, which is a kind of volatility spillover effect, and this relevance changes with time. Therefore, we chose the TVP-SV-VAR model to study the dynamic influence degree and time-varying characteristics of bank credit, house prices and carbon dioxide emissions. Then, the DCC-GARCH model was used to measure the dynamic correlation coefficient of bank credit, house prices and carbon dioxide emissions.

### 3.1. TVP-SV-VAR Model

We define the structural vector autoregression (SVAR) model:(1)$${\mathrm{Ay}}_{t}=F_{1}y_{t-1}+\ldots +F_{s}y_{t-s}+\mu_{t},\;t=s+1\ldots n$$
in which $y_{t}$ is the k×1 dimension vector of the variable to be observed, A, $F_{1},\ldots ,F_{s}$ are k×k dimension coefficient matrices, and disturbance term $\mu_{t}$ is k×1 dimension structural shock with $\mu_{t}\;\sim \;N\left(0,\Sigma \Sigma \right)$, where
$$\Sigma =\left(\begin{array}{cccc} \sigma_{1} & 0 & \cdots & 0\\ 0 & \ddots & \ddots & \vdots \\ \vdots & \ddots & \ddots & 0\\ 0 & \cdots & 0 & \sigma_{k} \end{array}\right)$$

Assuming A is a lower triangular matrix
$$A=\left(\begin{array}{cccc} 1 & 0 & \cdots & 0\\ a_{21} & \ddots & \ddots & \vdots \\ \vdots & \ddots & \ddots & 0\\ a_{k1} & \cdots & a_{k,k-1} & 1 \end{array}\right)$$
then Equation (1) can be rewritten as:(2)$${\;y}_{t}=B_{1}y_{t-1}+\ldots +B_{s}y_{t-s}+A^{-1}{\Sigma \varepsilon }_{t},\;\varepsilon_{t}\;\sim \;N\left(0,I_{k}\right),$$
where $B_{i}=A^{-1}F_{i}$, and i=1,2,…,s. We stack elements in $B_{i}$ into $k^{2}\times 1$ dimension column vector β, and define $X_{t}=I_{k}⊗\left(y_{t-1}^{\prime },\ldots ,y_{t-k}^{\prime }\right)$, where ⊗ refers to Kronecker product, therefore we obtain:(3)$$y_{t}=X_{t}\beta +A^{-1}{\Sigma \varepsilon }_{t},\;t=s+1\ldots n,\;$$

In Equation (3), all parameters are time-invariant ones. If they are transformed into time-variant parameters, the model is extended to a TVP-SV-VAR model [54,56].

Consider Equation (4):(4)$${\;y}_{t}=X_{t}\beta_{t}+A_{t}^{-1}\varepsilon_{t},\;t=s+1\ldots n$$
in which $\beta_{t},\;A_{t}$ and $\Sigma_{t}$ are all time-variant parameters.

Let $a_{t}={\left(a_{21},a_{31},a_{32},a_{41},\ldots ,a_{k,k-1},\right)}^{\prime }$, where $a_{t}$ is the stacked vector of lower triangular matrix $A_{t}$. Take $h_{t}={\left(h_{1t},\ldots ,h_{1t}\right)}^{\prime }$, in which $h_{\mathrm{jt}}=\log\left(\sigma_{\mathrm{jt}}^{2}\right)$, j=1,…,k, and t=s+1,…,n. Assume parameters in Equation (4) obey the random walk hypothesis:$$\beta_{t+1}=\beta_{t}+\mu_{\beta t}$$
$${\;\alpha }_{t+1}=\alpha_{t}+\mu_{\alpha t}$$
$$h_{t+1}=h_{t}+\mu_{\mathrm{ht}}$$
$$\left(\begin{array}{c} \varepsilon_{t}\\ \mu_{\beta t}\\ \mu_{\alpha t}\\ \mu_{\mathrm{ht}} \end{array}\right)\;\sim \;N\left(0,\left(\begin{array}{cccc} I & O & O & O\\ O & \Sigma_{\beta } & O & O\\ O & O & \Sigma_{\alpha } & O\\ O & O & O & \Sigma_{h} \end{array}\right)\right),t=s+1,\ldots ,n$$
where $\beta_{s+1}\;\sim \;N\left(\mu_{\beta_{0}},\Sigma_{\beta_{0}}\right)$, $\alpha_{s+1}\;\sim \;N\left(\mu_{\alpha_{0}},\Sigma_{\alpha_{0}}\right)$, and $h_{s+1}\;\sim \;N\left(\mu_{h_{0}},\Sigma_{h_{0}}\right)$.

### 3.2. Bayesian DCC-GARCH Model

Consider the GARCH model of multivariate time series ${y_{t}=\left(y_{1t},\ldots ,y_{\mathrm{kt}}\right)}^{\prime }$ as follows:$$y_{t}=H_{t}^{1/2}\epsilon_{t}$$
in which $H_{t}^{1/2}$ is the k × k arbitrary positive definite matrix, and $H_{t}$ is the conditional variance of $y_{t}$, which depends on finite parameter vector θ. The error vector obeys k×1 order independent and identical distribution. $E\left(\epsilon_{t}\right)=0$ and $E\left(\epsilon_{t}\epsilon_{t}^{\prime }\right)=I_{k}$, where $I_{k}$ is K order identity matrix.

We define the constant conditional correlation model:$$H_{t}=D_{t}{\mathrm{RD}}_{t}$$
where $D_{t}=\mathrm{diag}\left(h_{11,t}^{1/2},\ldots ,h_{\mathrm{kk},t}^{1/2}\right)$, and R is the positive definite symmetric matrix. The element in it is fixed-condition correlation coefficient $\rho_{\mathrm{ij}}$, i,j=1…k. When i=j, $\rho_{\mathrm{ij}}=1$. Each conditional variance is determined by $h_{\mathrm{ij},t}=\rho_{\mathrm{ij}}\sqrt{h_{\mathrm{ii},t}h_{\mathrm{jj},t}}$. Each conditional variance in $D_{t}$ is set as a monovariant GARCH model. We specify the GARCH (1, 1) model as follows:(5)$$h_{\mathrm{ii},t}=\omega_{i}+\alpha_{i}\varepsilon_{i,t-1}^{2}+\beta_{i}h_{\mathrm{ii},t-1},i=1\ldots k$$
where $\omega_{i}>0$, $\alpha_{i}\geq 0$, $\beta_{i}\geq 0$, $\alpha_{i}+\beta_{i}<1$, and i=1…k. The model contains k (k + 5)/2 parameters. If and only if $h_{\mathrm{ii},t}>0$, i=1…k, and R is a positive definite matrix, $H_{t}$ is a positive definite matrix.

By allowing the conditional correlation coefficient matrix to change with time, Engle (2002) proposed a more general CCC model, namely the DCC (dynamic conditional correlation) model [55]. Drawing on [55,57], in $H_{t}=D_{t}{\mathrm{RD}}_{t}$:(6)$${\;R}_{t}=\mathrm{diag}{\left(Q_{t}\right)}^{1/2}Q_{t}\mathrm{diag}{\left(Q_{t}\right)}^{1/2}$$
in which $Q_{t}$ is the k × k symmetric positive definite matrix.
(7)$$Q_{t}=\left(1-a_{i}-\beta_{i}\right)\;R+a_{i}\mu_{t-1}\mu_{t-1}^{\prime }+\beta_{i}Q_{t-1}$$
in which R is the unconditional covariance matrix of $\mu_{t}$, $\alpha_{i}\geq 0$, $\beta_{i}\geq 0$, $\alpha_{i}+\beta_{i}<1$ and i=1,…,k. $h_{\mathrm{ij},t}=q_{\mathrm{ij},t}\sqrt{h_{\mathrm{ii},t}h_{\mathrm{jj},t}}/\sqrt{q_{\mathrm{ii},t}q_{\mathrm{jj},t}}$.

The conditional likelihood function is:$$\ell \left(\theta \right)=\overset{n}{\underset{t=1}{\prod }}{\left|H_{t}\right|}^{-1/2}p_{\epsilon }\left(H_{t}^{-1/2}y_{t}\right)=\overset{n}{\underset{t=1}{\prod }}\left[\overset{k}{\underset{i=1}{\prod }}h_{\mathrm{ii},t}^{-1/2}\right]{\left|R_{t}\right|}^{-1/2}p_{\epsilon }\left({\left(D_{t}R_{t}D_{t}\right)}^{-1/2}y_{t}\right)$$
where $p_{\epsilon }$ is the joint probability density function of $\epsilon_{t}$. The parameter set of the model is $\theta =\left(\omega_{1},\alpha_{1},\beta_{1},\ldots ,\omega_{k},\alpha_{k},\beta_{k},\rho_{12},\ldots ,\rho_{k-1,k}\right)$.

The Bayesian method was used for this study. Due to the obvious thick-tail phenomenon in financial time series, when specifying distribution, GED distribution (a multivariate exponential power distribution), whose error term has skewness and thick-tail characteristics, is employed to estimate the DCC model. Its probability density function of the standard monovariant is:(8)$$p\left(x|\delta \right)={\left[\frac{\Gamma \left(3/\delta \right)}{\Gamma \left(1/\delta \right)}\right]}^{1/2}\frac{1}{2\Gamma \left(\left(\delta +1\right)/\delta \right)}\exp\left\{-{\left[\frac{\Gamma \left(3/\delta \right)}{\Gamma \left(1/\delta \right)}x^{2}\right]}^{\delta /2}\right\}$$

The kurtosis of the function derives from $\Gamma \left(1/\delta \right)\Gamma \left(5/\delta \right)/\Gamma \left(3/\delta \right){\;}^{2}-3$. The function presents a standard normal distribution when δ=2, a kurtosis distribution when δ<2, and has a thinner tail when δ>2. Bauwens and Laurent (2005) extended a multivariate model, whose marginal distribution and absolute moment are difficult to obtain [58], so the joint distribution of k independent random variables was used here. As a result, the marginal density of the above formula has a common tail parameter δ*,* whose joint density function is:(9)$$p\left(x|\delta \right)={\left[\frac{\Gamma \left(3/\delta \right)}{\Gamma \left(1/\delta \right)}\right]}^{k/2}\frac{1}{2\Gamma {\left(\left(\delta +1\right)/\delta \right)}^{k}}\exp\left\{-{\left[\frac{\Gamma \left(3/\delta \right)}{\Gamma \left(1/\delta \right)}\right]}^{\delta /2}\overset{k}{\underset{i=1}{\sum }}{\left|x_{i}\right|}^{\delta }\right\}$$

As Equation (8) is in a standardized form, it obeys E(X)=0, $\mathrm{Var}\left(X\right)=I_{k}$. Drawing on Bauwens and Laurent (2005) [58], we introduced an asymmetric form in multivariate distribution, namely GED $\left(0,\;I_{K},\;\delta \right)$. By calculating its first-order absolute moment $m_{1}=\Gamma \left(2/\delta \right)/\sqrt{\Gamma \left(1/\delta \right)\Gamma \left(3/\delta \right)}$, we obtained its mean value $\mu_{\gamma_{i}}$ and variance $\sigma_{\gamma_{i}}^{2}$. The standardized skewness form of GED is denoted as (0, $I_{k}$, γ, δ):(10)$$s\left(x|\gamma \right)=2^{k}\left(\prod_{i=1}^{k}\frac{\gamma_{i}\sigma_{\gamma_{i}}}{1+\gamma_{i}^{2}}\right)\;{\left[\frac{\Gamma \left(3/\delta \right)}{\Gamma \left(1/\delta \right)}\right]}^{k/2}\frac{\left\{-{\left[\frac{\Gamma \left(3/\delta \right)}{\Gamma \left(1/\delta \right)}\right]}^{\delta /2}\sum_{i=1}^{k}{\left|x_{i}^{*}\right|}^{\delta }\right\}}{{\left(2/\delta \right)}^{2}{\left[\Gamma \left(1/\delta \right)\right]}^{k}}$$
when $x_{i}\geq \mu_{\gamma_{i}}/\sigma_{\gamma_{i}}$, $x_{i}^{*}=\left(x_{i}\sigma_{\gamma_{i}}+\mu_{\gamma_{i}}\right)/\gamma_{i}$. When $x_{i}<\mu_{\gamma_{i}}/\sigma_{\gamma_{i}}$, $x_{i}^{*}=\left(x_{i}\sigma_{\gamma_{i}}+\mu_{\gamma_{i}}\right)\gamma_{i}$.

## 4. Empirical Analysis

### 4.1. Empirical Analysis Based on TVP-SV-VAR Model

For empirical analyses on relevant data from January 2000 to December 2019 conducted herein, data about bank credit and carbon dioxide came from the official website of the People’s Bank of China and CEADs (Carbon Emission Accounts Datasets), while the price of commercial housing was calculated based on the sales of commercial housing and the area of commercial housing released on the official website of China’s National Bureau of Statistics. We set the sequence variables in the model as bank credit (CM), house price (HP) and carbon dioxide (CO_2_), processed the TVP-SV-VAR model with Oxmetrics 6.2 and set MCMC sampling as 20,000 with the model lag period as 2.

#### 4.1.1. The Analysis of Parameter Regression Results

As shown in Table 1, the parameter estimation results of the model show that we can reject the null hypothesis of the Geweke test at the 5% significance level, and the maximum value of invalid influencing factors is 174.73. This indicates that the MCMC algorithm used in this study is effective for parameter estimation.

Figure 1 presents the auto-correlation coefficient, simulation path and posterior distribution of samples. It shows that after samples in the burn-in period are excluded, the auto-correlation coefficients of ∑_β_, ∑_α_, and ∑_h_ are all converged, which indicates the method used herein for value assignment on samples can effectively produce uncorrelated samples, and that the simulation is valid.

Figure 2 reflects the random fluctuation and time-varying characteristics of the structural shocks of bank credit, house prices and carbon dioxide emissions. It can be seen from the figure that the random volatility of the three variables shows consistency in time. Specifically, the random volatility of bank credit, house prices and carbon dioxide emissions has been at a high level in two time periods (2000 to 2003 and after 2017), while the random volatility between 2003 and 2017 is very small, even close to zero.

#### 4.1.2. Analysis of Time-Variant Impulse Responses

We can obtain two different shock response diagrams from the TVP-VAR model—those with different time points or different lead times. Figure 3 shows different time point impulse responses for December 2004, December 2009, and December 2014, respectively. Figure 4 shows different periods’ impulse responses with one-period-ahead, three-period-ahead and five-period-ahead.

(1)Analysis of the time-variant characteristics of impulse responses at different time points

As shown in Figure 3, although three shocks were applied at three completely different points in time, the change trend of the impulse response function is similar. Specifically, whether in December 2004, December 2009 or December 2014, house prices responded to the bank credit shock in the current period, rose rapidly, reached the maximum and rose to the highest point in the first period, and then descended rapidly to close to zero in period 6. Similar to the house price, the impulse response function change trend of carbon dioxide impact by bank credit at three different time points in December 2004, December 2009 and 2014 is similar. It reacted and reached the maximum in period two, and then gradually decreased. After period ten, the carbon dioxide response value in December 2004 and December 2009 changed from a positive range to a negative range, but the carbon dioxide response value of December 2014 was always positive.

The impulse response of bank credit and carbon dioxide emissions to house prices at three different time points was inconsistent. Specifically, for the house price shocks in December 2004 and December 2014, the response value of bank credit was always in the negative region, and decreased rapidly in the current period, then increased rapidly, then decreased rapidly again, and reached the minimum value in the third period. For the house price shock imposed in December 2009, the impulse response value of bank credit was in the negative region in the first four periods, but rose rapidly and become positive after the fifth period. The change trend of the impulse response function of carbon dioxide emissions to house price shocks at three different time points shows different characteristics. In particular, the impulse response result of carbon dioxide emissions to house prices in December 2009 is different from that in December 2004 and December 2014. For the house price shocks in December 2004 and December 2014, the impulse response of carbon dioxide emissions rapidly decreased to a negative value, and remained in the negative range. For the house price shock in December 2009, the response value of carbon dioxide emissions was always positive, and it responded in the current period, slightly decreased, fell to the minimum value in period two, and then showed a gradual upward trend.

The change trend of the impulse response function of bank credit and house prices on the impact of carbon dioxide emissions at three different time points is similar, and its response values are all positive. Among them, bank credit responds to the impact of carbon dioxide emissions in the current period and rises rapidly. The house prices respond to the current impact of carbon dioxide and rise rapidly, but the increase is smaller than that of bank credit. It means that, as a country at the stage of industrialization, in China, the increase of carbon emissions often means economic growth, which in turn will drive the rise of housing prices.

(2)Analysis of Time-Variant Characteristics of Impulse Response at Different Lead Times

As presented in Figure 4, whether it was one, three or five periods ahead, the impulse response trend of real estate prices to bank credit shocks was very similar. From the impulse response at one period ahead, the impact of bank credit on real estate prices was large and constantly positive. The impulse response value of carbon dioxide emissions to bank credit shock changed within the positive and negative range. From the impulse response at one period ahead and three periods ahead, the response value was positive in most of the time period, while from the impulse response at five periods ahead, the impact of bank credit on carbon dioxide emissions was negative before 2009, but positive after 2009. The impulse response value of bank credit to the impact of house prices was mostly negative. From the impulse response at one period ahead, the impact of house prices on bank credit was large and very stable. From the impulse response at one period ahead, the response of carbon dioxide emissions to house price shock varied between positive and negative values, which was negative in most of the time regions before 2009 and positive in most of the time regions after 2009. From the impulse response at one period ahead, the impulse response values of bank credit and house prices to the impact of carbon dioxide emissions were almost straight-line and constantly positive, which indicates that the impact of carbon dioxide emissions on bank credit and housing prices was relatively stable, among which the impulse response value of house prices was particularly large.

So far, we have analyzed in detail the internal action among bank credit, house prices and carbon dioxide emissions with the help of the TVP-SV-VAR model based on the realistic background of China’s economic transformation and upgrading with low-carbon energy conservation characteristics, policy preferences, etc. To make a further thorough inquiry about the dynamic relation among them, we measured the coefficients of their dynamic relation with the Bayesian DCC-GARCH model.

### 4.2. Empirical Analysis Based on Bayesian DCC-GARCH Model

#### 4.2.1. Estimation Results of the Model

The Bayesian DCC-GARCH model was used to analyze the dynamic relation among bank credit, house prices and carbon dioxide emissions (CO_2_). Table 2 presents the empirical results of the model.

In above table, parameter γ measures the skewness of bank credit, house price and carbon dioxide emissions. γ > 1 represents right-skewed, while γ ≤ 1 left-skewed. ω is a constant term of the variance equation of the model. α and β are coefficients of the ARCH term and GARCH term of the variance equation, respectively. The closer the sum α + β is to 1, the slower the decay rate of the volatility. υ measures whether the thick-tail distribution is applicable to the error term. a, b and their sum are used to detect whether the DCC model should be employed. If a + b = 0, the CCC model is applicable, otherwise it is more suitable. According to Table 2, parameter υ was greater than 2 in the mean regression model and the quantile regression model, which indicates the error term had a tailing effect. a + b = 0.7857~1.0203, and regression results of mean values and high quantiles were greater than 0.1. Therefore, it was reasonable to use the DCC model for analysis.

By observing the regression results, it is not difficult to find out that the volatility of bank credit, house prices and carbon dioxide emissions are asymmetric, with that of bank credit and carbon dioxide emissions obviously left-skewed, while that of house prices right-skewed. Being left-skewed means there is a higher probability for the volatility to fall to the right of the mean value, while being right-skewed indicates the probability of falling to the right of the average is relatively high. If volatility is regarded as a risk, being left-skewed is often considered as there are relatively large market risks behind it. The asymmetry of volatility shows the risk of volatility in bank credit and carbon dioxide emissions are relatively small, and that in houses price it is relatively large. In addition, the sums of α and β of the three sequences were all above 0.71. Table 2 indicates that, as the quantile increased, the value of α + β also increased. Specifically, at quantiles of 2.50%, 25.00%, 50.00%, 75.00% and 97.50%, the values of α + β of bank credit are 0.4144, 0.6754, 0.8299, 0.9781 and 1.2864, respectively, those of house prices are 0.3560, 0.6645, 0.8284, 0.9772 and 1.209, and those of carbon dioxide emissions are 0.7216, 0.8834, 0.9813, 1.0814 and 1.2422. To sum up, as the quantile continued to increase, the fluctuation of the sequences gradually intensified, and the attenuation speed of the fluctuation gradually slowed down.

#### 4.2.2. Analysis of Dynamic Correlation Coefficients

Figure 5, Figure 6 and Figure 7 show the dynamic correlation between bank credit and house prices, bank credit and carbon dioxide emissions (CO_2_) and house prices and carbon dioxide emissions (CO_2_) calculated with the Bayesian DCC-GARCH model. We draw the following conclusions from them:

Firstly, the coefficient of dynamic correlation between bank credit and house prices was always positive between 0.0545 and 0.6502. Seen from time, it shows an upward trend from 2001 to 2006 and a downward trend from 2007 to 2008, except for very few months, then rebounded rapidly from the beginning of 2009, and peaked in 2009, 2013 and 2018, respectively. As we all know, China’s house prices witnessed “spikes” in 2010, 2013 and 2018, which also confirmed bank credit is closely related to house prices, and that bank credit significantly promotes the rise of house prices. Afterwards, the coefficient of their correlation stabilizes and drops beginning from the second half of 2018, which indicates “financial deleveraging” gradually weakening their correlation. Secondly, the coefficient of dynamic correlation between bank credit and carbon dioxide was always negative, and fluctuated between −0.4372 and 0.6206 at an increasing amplitude. This shows that there was a positive correlation between the two variables in most sample periods, while there was also a negative correlation between them in some months of 2000, 2006, 2012 and 2015. That is to say, from the perspective of time, the surge in bank credit during the entire sample period aggravated the accumulation of carbon dioxide emissions with the exception of a few months in 2000, 2006, 2012 and 2015. Additionally, in the case of negative correlation between the two variables, the coefficient of correlation stands between −0.4372 and −0.0217, generally showing a upward trend. Thirdly, the coefficient of dynamic correlation between house prices and carbon dioxide emissions was always positive, varying between 0.0232 and 0.3589, except for some months in 2006, 2009, 2012 and 2015. That is to say, there was a positive correlation between these two variables for most of the time, which means rising house prices would probably exacerbate carbon dioxide emissions. In terms of time, the correlation showed a generally upward trend from 2000 to 2019. Especially from 2010 to 2019, the upward trend is very obvious. We can draw a clear and definite conclusion that with the flourishing growth of China’s property market, house prices rising rapidly in all likelihood intensifies the accumulation of carbon dioxide emissions to a certain extent, and that this promotive effect gradually manifested over time.

As revealed by Figure 5, Figure 6 and Figure 7, bank credit and house prices were most relevant, followed by bank credit and carbon dioxide emissions, then by house prices and carbon dioxide emissions. The dynamic correlations among the three elements were relatively stable: there was a positive correlation among bank credit, house prices and carbon dioxide emissions in most time periods, while there was a negative correlation between bank credit and carbon dioxide emissions, and between house prices and carbon dioxide emissions in very few time periods. This means that, currently, rise in bank credit will push up house prices and increase the accumulation of carbon dioxide emissions, moreover, the continued rise in house prices will also increase carbon dioxide emissions.

## 5. Conclusions and Recommendations

Continued rapid rises in China’s house prices and carbon dioxide emissions have attracted a growing amount of attention from scholars and policy makers. First, we constructed a TVP-SV-VAR model with both time-variant parameters and stochastic volatility and analyzed the mechanism of action and transmission among bank credit, house prices and carbon dioxide emissions in China in recent years. The results show that a rise in bank credit significantly pushes up house prices and carbon dioxide emissions, that the rise in house prices inhibits bank credit and exacerbates carbon dioxide emissions most of the time and that the increase in carbon dioxide emissions will stimulate the expansion of bank credit and the rapid rise of house prices. Second, we further explored the dynamic relation among bank credit, house prices and carbon dioxide emissions with the help of the DCC-GARCH model. We found there was a positive feedback mechanism or correlation among bank credit, house prices and carbon dioxide emissions in most time periods, and that bank credit and house prices are most relevant, followed by bank credit and carbon dioxide emissions, then by house prices and carbon dioxide emissions. It is obvious that the obtained results of this manuscript are consistent with some recent published papers which concern the real estate market and carbon emissions. For example, Ma et al. (2019) indicated factors such as housing purchasing power, housing price-to-income ratio and population per household play a crucial role in reducing carbon dioxide emissions [59]. Qashou et al. (2022) found out that the development of the real estate market probably has a profound impact on carbon dioxide emissions in Turkey [1]. In addition, the research on carbon reduction of the development in energy systems for residential applications are also worth mentioning. Some studies have shown that upgrading the household furnaces with high-efficiency models or substituting residential solar panel systems for fossil fuels are conducive to carbon emission reduction [60,61]. Prabatha et al. (2020) argued that building retrofitting can effectively improve energy performance in existing buildings [62]. The carbon capture technology (CCT) is a key technology to reduce carbon dioxide emissions, mitigate climate change and the negative impact on the environment [63]. Some scholars pointed out that installing carbon capture devices on the building floor is not only conducive to reducing carbon emissions but also conducive to increasing long-term economic benefits [64,65].

Currently, China is at a critical stage of a shifting period of growth rate, labor pains of structural adjustment, digestion period of early stimulus policies and economic transformation and upgrading, so it is crucial to hold relatively stable bank credit and house prices with low carbon dioxide emissions. However, the economic structure, fiscal and financial system, industrial structure, energy consumption structure dominated by coal, industrialization and urbanization process, population change and environmental policies together lead to the high bank credit scale, house prices and carbon dioxide emissions. Moreover, the sustained and rapid rise in property prices has resulted in the increase of debt leverage and financial pressure of the resident sector and overall macro-economy. All of these factors count against the transformation and upgrading of China’s economic growth. This paper shows that there is a close relationship among bank credit, house prices and carbon dioxide emissions. In particular, the surge in bank credit drives up house prices drastically and increases carbon dioxide emissions in most time periods. Therefore, we believe that in order to achieve healthy, green and sustainable economic growth in China, the government should fully consider the potential impacts of bank credit on house prices and carbon dioxide emissions while formulating financial policies. Especially when conducting deleveraging reforms on the real economy and the financial sector, it should strengthen the management of bank credit, comprehensively use monetary policies and macroprudential policies to effectively control bank credit, and thereby achieve stable, healthy, long-term and sustainable development of the macro-economy under the new normal state.

## Figures and Tables

**Figure 1 ijerph-19-10428-f001:**
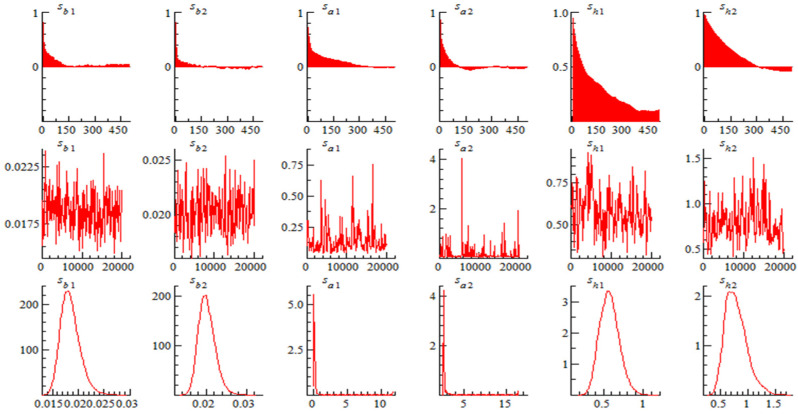
Estimation results of parameters in TVP-SV-VAR model.

**Figure 2 ijerph-19-10428-f002:**
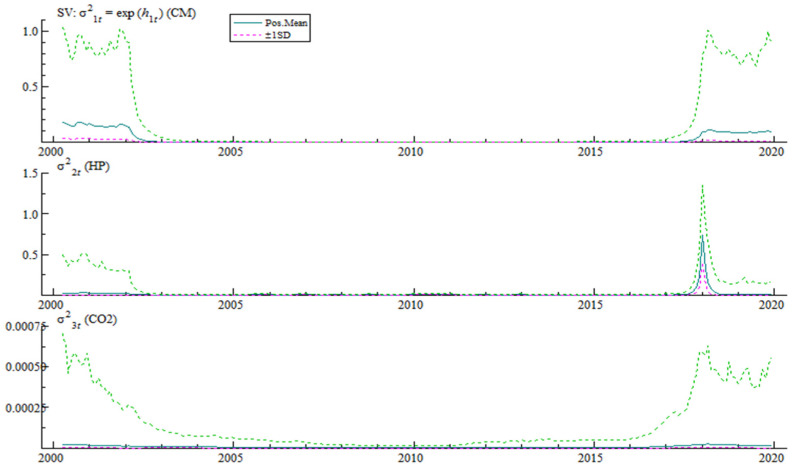
Random fluctuation and time-varying characteristics of structural impact.

**Figure 3 ijerph-19-10428-f003:**
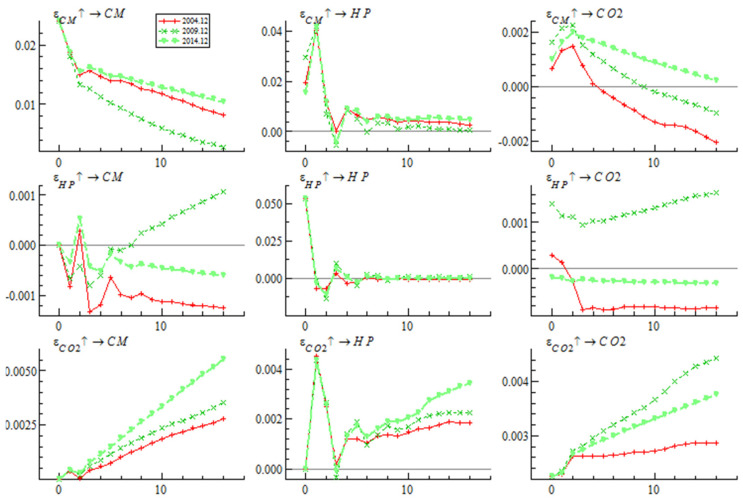
Impulse responses to shocks at different time points.

**Figure 4 ijerph-19-10428-f004:**
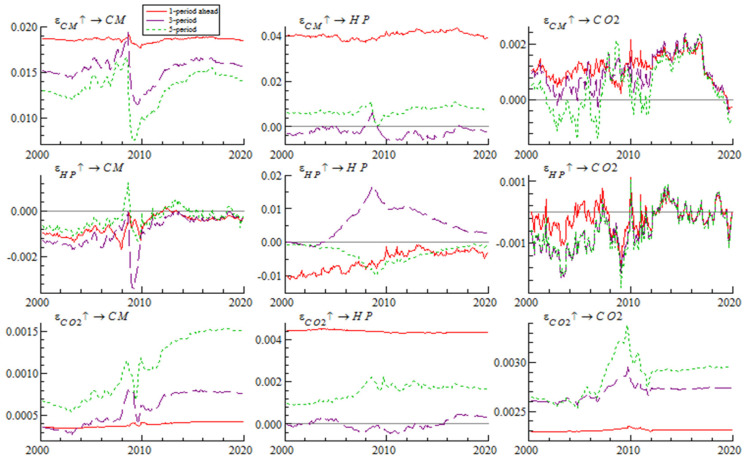
Impulse responses to shocks at different lead times.

**Figure 5 ijerph-19-10428-f005:**
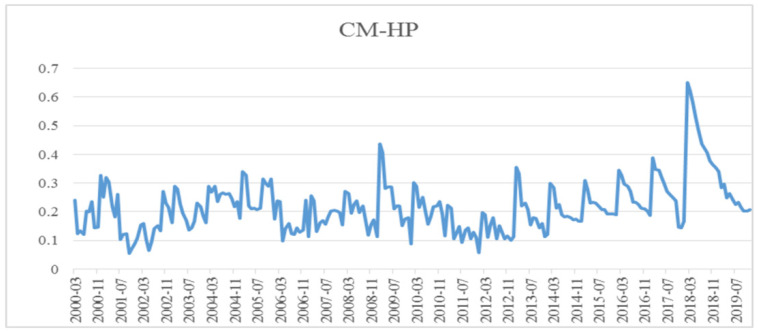
Diagram on the coefficient of dynamic correlation between bank credit and house prices.

**Figure 6 ijerph-19-10428-f006:**
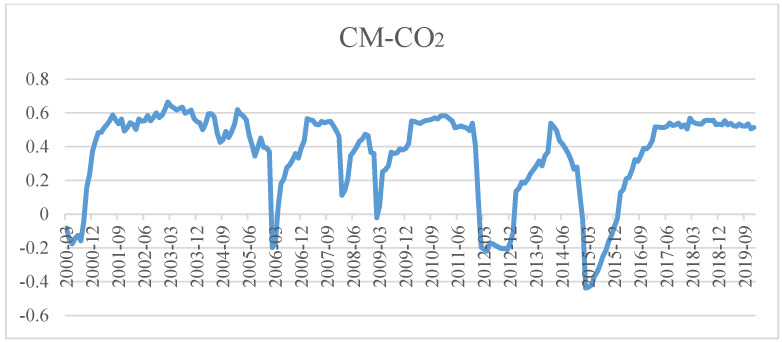
Diagram on the coefficient of dynamic correlation between bank credit and carbon dioxide emissions.

**Figure 7 ijerph-19-10428-f007:**
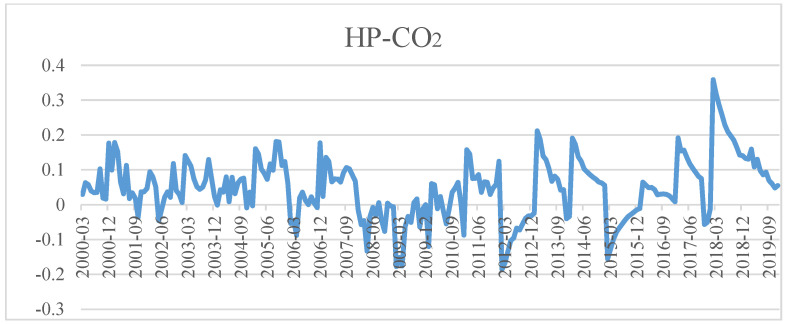
Diagram on the coefficient of dynamic correlation between house prices and carbon dioxide emissions.

**Table 1 ijerph-19-10428-t001:** Estimation results of TVP-SV-VAR model.

Parameter	Mean Value	Standard Deviation	95% Confidence Interval	Geweke	Non-Effective Factor
sb1	0.0185	0.0018	[0.0154, 0.0226]	0.061	45.17
sb2	0.0201	0.0020	[0.0166, 0.0246]	0.003	24.98
sa1	0.1469	0.1812	[0.0455, 0.4682]	0.373	74.61
sa2	0.2173	0.4888	[0.0384, 1.4980]	0.176	36.62
sh1	0.5750	0.1183	[0.3661, 0.8238]	0.476	166.35
sh2	0.7852	0.1928	[0.4762, 1.2344]	0.573	174.73

**Table 2 ijerph-19-10428-t002:** Monte Carlo estimation results of Bayesian DCC-GARCH model.

Variable	Parameter	Mean Value	Quantile				
			2.50%	25.00%	50.00%	75.00%	97.50%
CM	γ	0.5679	0.5197	0.5511	0.5695	0.5875	0.6216
	ω	0	0	0	0	0	0
	α	0.2998	0.1376	0.2294	0.2911	0.3571	0.5276
	β	0.5308	0.2768	0.446	0.5388	0.621	0.7588
HP	γ	1.168	1.049	1.12	1.162	1.211	1.318
	ω	0	0	0	0	0	0
	α	0.5832	0.2669	0.4843	0.5984	0.6944	0.8073
	β	0.2341	0.0891	0.1802	0.23	0.2828	0.4017
CO_2_	γ	0.5634	0.5071	0.5409	0.5625	0.5842	0.6228
	ω	0	0	0	0	0	0
	α	0.7014	0.5686	0.6527	0.6984	0.7519	0.8307
	β	0.2811	0.153	0.2307	0.2829	0.3295	0.4115
	υ	4.211	3.716	4.019	4.187	4.39	4.797
	a	0.0699	0.0389	0.0568	0.0685	0.0808	0.1105
	b	0.8372	0.7468	0.8113	0.8409	0.8667	0.9098

## Data Availability

Not applicable.

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
