# Peer review of "The Dynamic Relationship among Bank Credit, House Prices and Carbon Dioxide Emissions in China"

_ijerph, 2022, doi:10.3390/ijerph191610428_

Round 1

Reviewer 1 Report

From the abstract I got the impression the manuscript would compare consumption based GHG emissions - not territorial GHG emissions. This means I am less qualified (and less interested) to review the work. I also believe analyzing the relationship between bank credit - house price and consumption based emissions would be a very interesting article. 

Reviewer 2 Report

The Manuscript is very well written and explains the relationship between bank credits, carbon dioxide and house prices and provide sophisticated novel analysis of all three aspects simultaneously with clear conclusion which adds to comprehension of phenomenon affecting carbon emissions. As a result it should be accepted under minor revision.

The following improvements should be made to the manuscript.

1. In the introduction words such as "Nowadays", "Amazing","lots", etc. are used, please use the formal style of writing and rephrase sentences containing such words. There is also some mixture of American and British English, please choose one style for consistency.

2. Last paragraph in the introduction first two sentences should be combined as one, otherwise the first one is missing the "research question"  to which the word "Hence" is related.

3. It might be worth to substitute "a paper" to more formal "a manuscript" throughout the text.

4. Page 5 In the second sentence it is written as "green credit" please clarify that "green credit" is related to banks, as in some places it could be in form of government initiatives and tax reductions, sometimes known as "carbon credit" that was briefly discussed in e.g. Shukhobodskiy et al 2021 https://doi.org/10.1016/j.jclepro.2021.128926, e.g. “Green Financial Framework in the UK” https://www.gov.uk/government/publications/uk-government-green-financing#:~:text=Details,with%20clearly%20defined%20environmental%20benefits and e.g. Yeganeh et al. https://doi.org/10.1080/09613218.2020.1842165

5. Second sentence in the last paragraph of the literature review s. "In addition, they did not put bank credit, house price and carbon dioxide emissions into the same analysis framework or measurement model to make quantitative analysis about the mutual influence or dynamic relationship among them." please rephrase this sentence to more moderate one, for example "mentioned manuscripts in the literature review, have not combined..."

6. Please improve the quality of mathematical formulas, for example in the Equation 4 subscripts "\tau" are not on the same level, although symbols are, same happens in the the sentence that follows that formula.

7. In equation 7 \alpha and \beta have no subscripts however, later in the text both have subscripts in the definition, please make it clearer if in case both are the same or both are different. In case they are the same please rewrite symbols for consistency.

8. If it is possible please add some information in conclusion of how the obtained results relate to the situations in other countries. Moreover, the recent development in energy systems for residential applications have significant impact on carbon reduction, it might be worth to mention in conclusion, that presented trend might change due to technological advancements and regulations. For example in the EU and the UK, the non-efficient energy systems such as those with gas boilers are phased due to mandatory regulations or financial leavy to fully electric ones such as heat-pumps, storage heaters or combination of elements such as hybrid energy systems.

Reviewer 3 Report

This study analyzed the mutual relationship among bank credit, house price and carbon emissions in the context of China, which is an interesting topic. But the study lacks solid theoretical basis, therefore, I don’t think it is qualified for publication. 

Comments:

1.     Generally, when discussing the relationship among bank credit, house price and carbon emissions, there should be at least six relationships between any of these two. But the current literature review only mentions three of them, which is insufficient. Moreover, I have two main questions regarding this part. First, Section 2.3 fails to support the statement that house price has impacts on carbon emissions. The theoretical rationale behind the statement that house price can promote or inhibit the carbon emissions is not clear to me. Second, there is no statements about the impacts of carbon emissions on house price or bank credit, which give us no clues why the change of carbon emissions can affect the change of price or credit. 

2.     Qashou et al, 2022 mentioned that the real estate market has influence on carbon emissions, instead or house price. So the statement of “Some studies have pointed out that … through two channels…” in Introduction section is not correct. 

3.     The spatial scale adopted in this study is not clear, which is a major limitation of the results. At different spatial scale, the drivers of house price or bank credit are different. Some of them may cause the change of carbon emissions, while the others will not. For example, at the micro scale, house price may go up due to the newly-built metro station nearby, but the increment of house price in such case may not affect carbon emissions. But if this study is carried at the national scale, bank credit may be more associated with the policy, which also has no significant and direct relationship with carbon emissions. More justification is needed.

4.     The methodology section should be rewritten. First, the current version only introduces the models in general, but doesn’t explain how the authors applied these models. Second, why these two models are selected should be justified, too.

5.     Why does carbon emissions have positive impact on the house price and bank credit? This findings seems surprising but lack of enough discussion and explanation in this study. 

Author Response

Pleaase see  the attachment.
